# An Interdisciplinary Assessment of Biochemical and Antioxidant Attributes of Six Greek *Vicia sativa* L. Varieties

**DOI:** 10.3390/plants12152807

**Published:** 2023-07-28

**Authors:** Eleni D. Myrtsi, Dimitrios N. Vlachostergios, Christos Petsoulas, Epameinondas Evergetis, Sofia D. Koulocheri, Serkos A. Haroutounian

**Affiliations:** 1Laboratory of Nutritional Physiology and Feeding, Department of Animal Science, School of Animal Bioscience, Agricultural University of Athens, Iera Odos 75, 11855 Athens, Greece; elenamirtsi@aua.gr (E.D.M.); epaev@aua.gr (E.E.); skoul@aua.gr (S.D.K.); 2Institute of Industrial and Forage Crops, Hellenic Agricultural Organization ELGO-DIMITRA, 41335 Larissa, Greece; petsoulaschristos@elgo.gr

**Keywords:** *Vicia sativa* L., vetch, fatty acids, phenolic compounds, flavonoids, antioxidants

## Abstract

Common vetch (*Vicia sativa* L.) is one of the most cultivated feed crops with extensive agricultural diversity and numerous cultivars. This study concerns the first-time investigation of the dry plant biomass and grains of six vetch cultivars to define the detailed fingerprint of their phenolic and fatty acid content, along with their respective antioxidant potencies. The results revealed a substantial variation in the feed quality traits among the tested *Vicia sativa* varieties, highlighting the crucial role and influence the genotype plays in the achievement of high-quality livestock nutrition. Among the six varieties tested, *Istros* and *M-6900* displayed a particularly intriguing phytochemical profile characterized by elevated phenolic content, significant antioxidant potency and remarkably high fatty acid indices. These findings are indicative of the great potential of these varieties to function as suitable candidates for incorporation into farm animal diets either in the form of dry biomass (hay) or as a grain feed additive.

## 1. Introduction

The biochemical content in plants constitutes a definitive factor of their nutritional value. Various primary metabolites such as starch, proteins, sugars and fats have been utilized for the regulated description [1,2,3,4] and classification [5,6] of foods or the characterization of the respective agricultural production [7]. Among these biomolecules, fatty acids constitute a distinctive structural cluster with well-defined nutritional value [8,9] and well documented bioactivities, including anticancer, antiatherogenic and anti-inflammatory properties [10]. Although the chromatography-assisted determination of fatty acids, which are commonly referred to as the total fatty acid content, is a wider used methodology applied for the exploitation of their nutritional properties [8,11], there are only limited reports on the detailed determination of the fatty acids’ presence in plant tissues.

Recently, the assessment of content of secondary metabolites in plant tissues has also attracted considerable research attention as an intriguing topic inspired by their incorporation in various pharmaceuticals [12], cosmetics [13], foods [14] and feed [15]. Among the diverse categories of natural products involved in a plethora of industrial applications, phenolic compounds comprise the most frequently exploited class of compounds since they display numerous health beneficial effects and are being used as active ingredients in antioxidant [16,17], anti-inflammatory [17,18] and anticancer [19] preparations. Among various classes of phenolic compounds, flavonoids constitute the most pronounced structural cluster with well documented bioactivities [20] due to their antioxidant [21], anticancer [22] and antibacterial [23] properties. Although an inclusive chromatographic assessment constitutes the most commonly utilized methodology for the determination of phenolic [24,25] or flavonoid [26,27] content and the definition of their nutraceutical properties, there are only scarce reports concerning the determination of their molecular diversity which may be utilized for the cumulative estimation of their bioactivity. The identification of these two distinctive knowledge gaps defines the primary target of the present work, which aspires to delineate the relationship between the phytochemical content (fatty acids and phenolics) and bioactivity of plants and their genetic and environmental diversity.

The selection of the investigated plant species was based on a literature survey concerning the phenolic and/or fatty acid content in numerous plant-originated foods [28,29,30,31,32] and feeds—fodders [33,34]—which highlighted the common vetch (*Vicia sativa* L.) as an intriguing target. The common vetch is a feed crop expanded globally through its extensive agricultural diversity and numerous cultivars [35,36]. Accordingly, vetch comprises the main forage crop of Greece that is cultivated in rainfed fields for grain, hay, grazing, green manure or intercropping with cereals [37,38]. Its cultivation occupies an area of 22,954 ha for hay production, 36,861 ha for grain production and 7903 ha for grazing, yielding 4.12 tons/ha of hay and 1.8 tons/ha of grain [39].

This study concerns the exploitation of six vetch cultivars and reports for the first-time the detailed fingerprint of the content of their phenolic and fatty acid compounds, along with their antioxidant potencies. The final goal is to present and discuss the usefulness of secondary metabolite fingerprinting in the indication and/or identification of a plant’s biochemical functions in given conditions. 

## 2. Results and Discussion

### 2.1. Phytochemical Analyses

The applied solvents and extraction yield of the samples studied are summarized in Table 1, defining methanol as the most potent extraction solvent. Among the samples studied, the *BI-233* variety displayed the best overall extraction yields, followed by the *Istros* variety. 

#### 2.1.1. Total Tannins

The presence of tannins is generally considered as an antinutritional factor. Thus, there are extensive studies and literature reports concerning the tannin content in numerous *Vicia* sp. *taxa*, which reveal that tannins are mainly present in their grain extracts in quantities ranging from 0.61 to 1.39 %w/w as leucocyanidin equivalents [40]. In another study concerning the tannin content in the *V. sativa* grains, the TTC value was 1660 mg CE/Kg DM [41], which is in line with our reported value for the *BI-65* grains (1603.16 mg CE/Kg DM). 

All TTC values determined in this study are presented in Table 2, indicating that tannins were not detected in the grain extracts of the *Istros* and *M-6900* varieties. On the contrary, the *BI-65* grains were found to contain the highest tannin concentration among all the extracts studied. With respect to the plant tissues, tannins were not detected in the extracts of the *BI-233*, *Leonidas* and *Istros* plants, while the extracts of the *Line A*, *BI-65* and *M-6900* varieties contained large amounts of tannins (680.53, 408.67 and 876.27 mg CE/Kg DM, respectively). In a similar study concerning the common vetch varieties grown in different environments, the TTC mean values of plant biomass ranged from 350 to 370 mg CE/Kg DM [41,42]. It is noteworthy however that the extracts of both plant biomass and grains of the *BI-65* variety were found to contain large amounts of tannins, indicating this plant’s potential for incorporation into animal feeds, as a bioactive ingredient, due to the established antimicrobial, antiparasitic, antiviral, antioxidant and anti-inflammatory properties of tannins [43].

#### 2.1.2. Total Phenolics

Phenolic compounds are considered potent nutraceuticals. Their presence in various *Vicia* sp. *taxa* grains has been repetitively determined ranging from 7.72 to 16.42 mg/g, with the *V. sativa* grains displaying the lowest phenolic content [40]. Another study concerning the phenolic content in the grains of five different vetch genotypes indicated that their TPC values were comparable, displaying an average value of 2175 mg GAE/Kg DM [44]. On the other hand, the TPC values for the vetch grains determined in this study were substantially lower (420.2 to 656.4 mg GAE/Kg DM). 

It must be noted that all previous studies on vetches’ phenolic contents concerned mainly the extracts of the *V. sativa* grains. Thus, the current comparative study of plant biomass vs. grains represents the first thorough exploitation of vetches’ phenolic content. The results of the present study are included in Table 2, indicating that all the investigated varieties can be considered as potent sources of phenolic compounds. They also establish that vetches’ plant biomass contains considerably larger amounts of phenolics as compared to the respective grains. The varieties with the richest phenolic content were *M-6900*, *Istros* and *BI-233* having the TPC values of 1851.3, 1610.2 and 1418,3 mg GAE/Kg DM, respectively. These results are indicative of the suitability of the plants’ biomass for incorporation into animal rations as a source of phenolic compounds. The latter are known to exhibit potent antioxidant activities, along with beneficial properties for animal gut health, such as anti-inflammatory or antimicrobial activity. 

#### 2.1.3. Fatty Acids Fingerprint 

The fatty acid fingerprinting results obtained in this study constitute a significant progress towards the exploitation of the dry biomass and grains of the six varieties of the common vetch for utilization as forage. Previous reports established that *Vicia* sp. *taxa* synthesize a broad variety of fatty acids, with the palmitic, oleic, linoleic and linolenic acids being most abundant [45]. It is noteworthy however that most of these reports (96 publications) concern only their grains’ content, while reports on the fatty acid content in dry biomass are limited to only eight studies. Thus, the present work represents the first report containing a thorough fingerprinting of fatty acid content in the common vetch plants and the first-time quantitation of the presence of 27 fatty acids. Among them, the presence of caproic, capric, tridecanoic, pentadecanoic, *cis*-10-pentadecenoic, margaric, *cis*-10-heptadecenoic, *cis*-11-eicosenoic, *cis*-11,14-eicosadienoic, heneicosanoic, *cis*-11,14,17-eicosatrienoic, *cis*-5,8,11,14,17-eicosapentaenoic, tricosanoic, nervonic, erucic and lignoceric acids is reported for the first time in the vetch’s biomass. The same molecules, except those of erucic and lignoceric acids, were also detected in the grain studied. 

Table 3 summarizes the results of the fatty acid determinations, identifying the presence of 27 molecules in various concentrations. Specifically, the results of our study indicate that the fatty acid content in the *Istros* variety’s biomass exceeded by almost 500% the respective content in the other varieties investigated, while its grains also contain almost 25% more of fatty acids as compared to the other varieties studied. Specifically, the *Istros* variety displayed the highest content for almost all detected fatty acids, both in biomass (6872.45 mg/Kg DM) and grain (1948.80 mg/Kg DM) extracts. It is notable that the fatty acid content in its biomass was almost six times higher as compared to varieties *Line A* (1211.04 mg/Kg DM), *BI-65* (1095.17 mg/Kg DM) and *Leonidas* (1107.59 mg/Kg DM) and almost four times higher than that in varieties *BI-233* (1766.68 mg/Kg DM) and *M-6900* (1588.23 mg/Kg DM). 

With respect to the individual fatty acid content, previous reports have already identified as the prevailing fatty acids the palmitic, linoleic, α-linolenic and oleic acids [35,46]. Additionally, the myristic, palmitoleic, stearic, eicosanoic, erucic and lignoceric acids have also been detected in vetch grains [46], while the lauric, myristic, palmitic, palmitoleic, stearic, eicosanoic, oleic, linoleic and α-linolenic acids were identified as components of vetch biomass. 

It is noticeable that the fatty acid fingerprint may serve as a biomarker of vetch varieties. Specifically, among the fatty acids detected in the samples studied, *cis*-10-pentadecenoic acid was detected only in the *BI-65* biomass, while the molecules of *cis*-10-heptadecenoic acid, *cis*-11,14,17-eicosatrienoic acid and *cis*-5,8,11,14,17-eicosapentaenoic acid were found only in the biomass of the *BI-233* and *Leonidas* varieties, indicating a possible genetic similarity. On the other hand, the caproic and capric acids which were uniformly found in the biomass of all varieties were absent in their grain extracts, indicating the potential utilization of these molecules as biomarkers for the characterization of biomass origin and purity. 

#### 2.1.4. Phenolic Compounds Fingerprint

The fingerprinting results for phenolic compounds presented in this study constitute the first detailed determination of the phytochemical content in the common vetch. The phenolic compounds’ fingerprinting includes the exploitation of 67 phenolics, which led to the identification of the presence of 23 phenolic molecules in the samples studied. The results are presented in Table 4. It is noticeable that previous studies mostly focused on the presence of only a limited number of phenolics, identifying the presence of apigenin, kaempferol, quercetin [47] and the caffeic, p-coumaric, ferulic and sinapic acids [48] as the constituents of the common vetch. Our study reports for the first-time the identification of the presence of the following components as *V. sativa’s* constituents: chlorogenic acid, procyanidins B1 and B2, quercetagetin-7-O-glucoside, gallocatechin, secoisolariresinol, epigallocatechin gallate, 3′,4′,7-trihydroxyisoflavone, ononin, isoliquiritigenin, daidzein, glycitein, genistein, daidzin, glycitin, rutin, isoquercetin, hesperidin and quercitrin. On the other hand, the caffeic, gallic, neochlorogenic, p-coumaric, protocatechuic, sinapic acids and the molecules of catechin, epicatechin, epigallocatechin, eriodictyol, hesperetin, isorhamnetin, kaempferol, liquiritigenin, liquiritin, luteolin-4′-O-glucoside, myricetin, calycosin, pelargonidin, pelargonin, quercetagetin, rhamnetin, taxifolin, 4′,6,7-trihydroxyisoflavone, biochanin A, calycosin-7-O-D-glycoside, daidzein-7-O-glucuronide, equol, formononetin, genistein-7-O-glucuronide, genistin, puerarin, sissotrin, sophoricoside, phloretin, phloridzin, xanthoxumol, lariciresinol, matairesinol, coumestrol, hydroxytyrosol, polydatin and resveratrol, which were also included in the fingerprinting endeavor but were not detected in any of the examined extracts. 

Specifically, six major classes of flavonoid compounds were detected in the extracts studied. In particular, among the detected phenolic acids, the molecule of chlorogenic acid was found to be a component in all the tested varieties, except *Istros*. For lignans, the molecules of secoisolarisiresinol were present in all the varieties investigated except *M-6900*, while for chalcones, the molecules of isoliquiritigenin were present in all varieties. The isoflavones glycitein, genistein, daidzein and 3′,4′,7- trihydroxyisoflavone were detected in all the varieties studied, except glycitein, which was not present in *Line A*, genistein, which was not present in *Leonidas,* and the molecules of daidzein and 3′,4′,7-trihydroxyisoflavone which were found only in *M-6900*. The flavanones apigenin, luteolin and diosmetin were present in all varieties, except *BI-233,* which did not contain apigenin and luteolin, Line A, which did not contain luteolin, and BI-65, which did not contain diosmetin. Finally, the flavonols isoquercetin, rutin and quercetagetin-7-O-glucoside were present in all varieties, except quercetagetin-7-O-glucoside, which was not detected in the *BI-233* variety. 

Among the 24 individual flavonoids found in the investigated extracts, the molecules of isoquercetin (flavonol) and isoliquiritigenin (chalcone) were present in all extracts, while the flavonols isoquercetin and ononin were detected in all grain extracts except M-6900. We noticed however that the most abundant phenolic compound was the lignan secoisolariresinol, which was the prevailing component of the plant biomass of the Istros, BI-233 and Leonidas varieties. Flavonoid isoquercetin was the prevailing phenolic compound of the BI-65, M-6900 and Line A biomass. On the contrary, the respective grain samples displayed a particularly poor content of phenolics with the molecules of chlorogenic acid constituting the main component of the BI-65 grains.

### 2.2. Antioxidant Activity Assessment 

Previous literature reports provided the DPPH results of the antioxidant capacity assessment of 80:20 ethanol:water extracts of *V. sativa* and soybean seeds, indicating as more active the *V. sativa* seeds [49], while the antioxidant activity of their aerial part was investigated using assays to measure the degradation of various radicals, such as DPPH, superoxide anion and peroxyl ion [47]. In this study, the antioxidant capacities of the investigated samples were estimated using two complementary methods, the DPPH and FRAP bioassays. The assessment results are summarized in Table 5, indicating that in the DPPH assay, the highest antioxidant activity was detected in the methanolic extracts, while in the FRAP assay, both methanol and dichloromethane extracts displayed high antioxidant activity. 

Specifically, the DPPH assay results showed that the dichloromethane extracts of both plant biomass and grains were inactive, while the respective methanolic extracts were active except in the *BI-233* variety sample. Finally, the hexane extracts displayed weak DPPH antioxidant activity, with higher activity in the extracts of the *M-6900* and *Istros* varieties. Among the varieties tested, the methanolic extracts of plant biomass of the *M-6900* and *BI-65* varieties were determined as the most potent with1609.9 and 611.3 mg TE/Kg DM, respectively. The FRAP assay revealed that the dichloromethane extract of the *Istros* biomass was the most effective (34.02 mmol Fe (II)/Kg DM), followed by the methanolic extract of the *BI-65* biomass (25.02 mmol Fe (II)/Kg DM). 

With respect to the grain extracts, it must be noted that Orak et al. [50] previously studied the antioxidant activity of the methanolic extracts of grains obtained from different genotypes of *Vicia* species, using the DPPH, ABTS and FRAP assays. Their results indicated that the extracts of *V. sativa* genotypes exhibited higher antioxidant activity in the ABTS and FRAP assays. Specifically, the antioxidant activity of their extracts in the FRAP assay ranged from 8.5 to 26.6 mmol Fe (II)/Kg DM. Our FRAP values for the methanolic extracts of the grains in this study are comparable with those for the dichloromethane extracts, presumably because the extraction with this solvent preceded the extraction with methanol, allowing the majority of the antioxidant compounds to be extracted into this solvent. Thus, the dichloromethane extract of the *BI-65* grains was the most effective (15.99 mmol Fe (II)/Kg DM), followed by the respective extract of the *Istros* variety (8.77 mmol Fe (II)/Kg DM). Finally, in the DPPH assay, the methanolic grain extracts of the *BI-65* and *Istros* varieties displayed the antioxidant activity at 152.3 and 85.0 TE/Kg DM, respectively. 

### 2.3. Diversity of Vicia Sativa Genotypes Based on Phytochemical and Antioxidant Descriptors

#### 2.3.1. Hierarchical Cluster Analysis

A hierarchical cluster analysis of the phytochemical and antioxidant traits, as well as the different genotypes of *Vicia sativa,* was performed applying the Ward’s method. The resulting clusters are illustrated in Figure 1.

Two dendrograms were generated, one for the varieties and another for the descriptors. Among the varieties, three distinct groups were identified in the first dendrogram. Cultivar *Istros* emerged as a separate group, while *M-6900*, *BI-233*, *Leonidas* and *BI-65* formed another group. The last group consisted of *Line A* and the grain of *Istros*, *M-6900* and *BI-65*. Additionally, a heatmap was created to visualize the values of the quality traits. This heatmap clearly highlights the superiority of the *Istros* variety. Finally, their quality traits are classified into two main groups. The first group comprises margaric and linolenic fatty acids, while the second group includes all the remaining descriptors.

#### 2.3.2. Principal Component Analysis (PCA)

Figure 2 showcases plots illustrating the contribution of the phytochemical and antioxidant quality traits, along with the different genotypes of *Vicia sativa*, to the first two components of the analysis. These plots were generated as part of the principal component analysis (PCA) performed to decrease the dimensionality of the original variables and aid in the identification of patterns, relationships and variations within the data.

The PCA was capable of highlighting the importance of the quality traits used in the analysis. Apart from margaric and linolenic fatty acids and the FRAP/MeOH antioxidant assay, all other descriptors exhibited a strong correlation and made nearly equal contributions to the first component, which accounted for 75.4% of the variability. The cumulative effect of the first two dimensions of the analysis explained a significant 86.3% of the overall variation. It must also be noted that PCA revealed an additional pattern that indicate the substantial contribution of the Istros variety to the first two components (>70%). This clear differentiation separates Istros from the other varieties, indicating its distinctive characteristics.

## 3. Materials and Methods

### 3.1. Materials

All *Vicia sativa* varieties examined were provided by the Institute of Industrial and Forage Crops (IIFC) of the Hellenic Agricultural Organization “ELGO-DIMITRA”. The genetic material studied was consisted of one advanced line (Line A) and five commercial varieties developed by the *Vicia sativa* breeding program of IIFC [51]. Plants were grown in field plots at the central farm of IIFC in Larissa (39°36′ N, 22°25′ E) during the culture period 2020–2021. The sampling of fresh biomass was performed at the pod setting stage (near 50% of the pods had reached their final length). This is considered the appropriate biomass stage for use as feed. A sample of 0.5 m^2^ was collected to ground level with manual shears throughout each plot, following by a visual guided 20 to 30 collection schemes that included proportionally all observed diversity of the field. The plant material was weighted fresh, then oven-dried for 48 h at 60 °C before subjecting it to analysis. Grain samples of *BI-65*, *Istros* and *M-6900* cultivars were obtained at their maturity level by the hand-harvesting of 1 Kg and subsequent threshed using a laboratory thresher (Wintersteiger LD350).

### 3.2. Chemicals and Standards 

Analytical grade hexane, dichloromethane and methanol were obtained from Fisher Chemicals (Hampton, NH, USA) and used as extraction solvents. LC-MS grade solvents, water and acetonitrile were purchased from J.T. Baker (Phillipsburg, NJ, USA), while formic acid (LC-MS grade) was obtained from Fisher Chemicals. For the LC–MS/MS determinations (Hampton, NH, USA).

All standards used for the phytoestrogens’ quantification analysis were provided by ExtraSynthese (Genay, France), except equol and puerarin, which were obtained from TCI (Tocyo Chemical Industry, Zwijindrecht, Belgium), and calycosin, calycosin-7-O-β-D-glucoside, matairesinol, lariciresinol and secoisolariciresinol, which were purchased from Biosynth Carbosynth (Compton, United Kingdom). The purity of al standards was >95%, except 3′,4′,7-trihydroxyflavone (purity > 90%). The 2-(4-chlorophenyl) malonaldehyde molecules used as internal standard were obtained from Sigma-Aldrich (Burlington, MA, USA)). The standard used for the fatty acids’ quantification analysis was the Supelco 37 Component FAME Mix from Sigma-Aldrich.

Potassium hydroxide (KOH) and boron trifluoride (BF_3_) (14% in methanol) were obtained from Sigma-Aldrich, and hydrochloric acid (HCl) was from Fluka (Buchs, Switzerland).

### 3.3. Extraction 

Immediately after their collection, all varieties of *Vicia* plants were placed in a dark and well-ventilated area and allowed to dry. The dried material was powdered and extracted successively with *n*-hexane, dichloromethane and methanol according to the following protocol: Samples of various weights, ranging from 30 to 100 g presented in Table 1, were extracted using the 1:7 ratio by analogy with the previous solvents for 48 h. Each solvent extraction was performed in triplicate and the combined solvent extracts were evaporated under vacuum using a rotary evaporation in temperatures ranging 25–35 °C. (Büchi Rotavapor R-210 equipped with Büchi vacuum pump V-700, Vacuum controller V-850 and Julabo F12 cooling unit). The hexane extracts were used for the determination of fatty acid content and the methanolic extracts were used for the determination of phenolic content.

### 3.4. Total Phenolic Content (TPC)

The TPC determination was performed through a well-established spectrophotometric method [52], adapted to the present study focus. In summary, 10 μL of methanolic extract were mixed with 100 μL of water and Folin–Ciocalteu reagent solution (10 μL) was added in a 96-well microplate in triplicate (Sarstedt AG & Co. KG, Nümbrecht, Germany). After incubating this mixture for 3 min at room temperature, 20 µL of 7.5% w/v sodium carbonate aqueous solution and 60 μL of water were added, followed by 60 min incubation in the dark. Then, the absorption was measured at 765 nm wavelength, and the TPC was determined against a previously prepared gallic acid standard calibration curve. The results are expressed as mg of gallic acid equivalents per Kg of dry material (mg GAE/Kg DM). 

### 3.5. Total Tannin Content

TTC was determined in accordance with a modification of a previously reported [52] spectrophotometric method. In particular, 25 µL of methanolic extract was mixed with 150 µL of a 4% *w*/*v* vanillin methanolic solution, and an equal volume of 32% methanol solution of sulfuric acid was added. The mixture was placed into a 96-well microplate in triplicate and incubated at room temperature in the dark for 15 min. The absorbance of each solution was measured at 500 nm wavelength, and the TTC value was determined against a standard calibration curve of catechin. The results are expressed as mg of catechin equivalents per Kg of dry material (mg CE/Kg DM). 

### 3.6. Determination of Phenolic Compounds Content by LC–MS/MS Analysis

The determination of phytoestrogens content was performed using an Accela ultra-high-performance liquid chromatography system coupled with a TSQ Quantum Access triplequadrupole mass spectrometer (Thermo Fisher Scientific, Inc., Waltham, MA, USA) as previously described. For the determination of phenolics a C18 column (150 × 2.1 mm, 3 μm, Fortis Technologies Ltd., Neston, Cheshire, UK) coupled with an AF C18 guard column (10 × 2.0 mm, 3 μm, Fortis Technologies Ltd., Neston, Cheshire, UK) was used. The column was eluted with mobile phase A (water with 0.1% formic acid) and mobile phase B (100% acetonitrile). The flow rate was set at 280 µL/min, and the gradient elution conditions were adjusted for the re-equilibration of the column between the injections as follows: 0.0–2.0 min, 20% B; 2.0–25.0 min, from 20% to 51% B; 25.0–30.0 min, 51% to 70% B; 30.0–30.1 min, 20% B; 30.1–35.0 min, 20% B. The sample temperature was set at 25 °C, and the column temperature at 32 °C. The sample injection volume was 10 μL. 

The MS/MS was conducted using the electrospray ionization (ESI) technique in the selected reaction monitoring (SRM) mode. The capillary temperature was set at 300 °C, using nitrogen as a sheath and auxiliary gas supplied by a nitrogen generator (Peak Scientific, Glasgow, UK). The initial gas pressures were set at 35 and 10 Arb, respectively. The collision pressure of the argon gas was adjusted at 1.5 m Torr, and the spray voltage was set at 3.5 kV in both positive and negative polarities [53].

### 3.7. Determination of Fatty Acids Content by GC-FID Analysis

Prior to analysis, it is necessary to esterify the fatty acids. For this purpose, a method described in [54] was used with some modifications. Specifically, 50 mg of hexane extract was hydrolyzed with the addition of 1 mL of 1 M KOH in 70% ethanol at 90 °C, in a block heater for 1 h. Then, the reaction mixture was acidified with the addition of 1.2 mL 1 M HCl, and the mixture was vortexed. Into this mixture, 1 mL of hexane was added, stirred for 1 min, the hexane phase was separated and evaporated under vacuo (T < 25 °C). To the remaining slurry that contained the methylated fatty acids, a solution of 1 mL 14% BF_3_ in methanol was added, and the mixture was stirred for 20 min at 37 °C. The resulting solution was quenched with the addition of water, and the fatty acid methyl esters were extracted with 1 mL hexane and subjected to the analysis.

The analyses of fatty acids were performed using a 7820A GC-FID System (Agilent Technologies, Inc., Santa Clara, CA, USA), using Hydrogen (H_2_) from a hydrogen generator (Peak Scientific, Glasgow, UK), nitrogen (N_2_), used as make-up gas and synthetic air, both of high purity. The column was DB-WAX Ultra Inert 30 m, 0.25 mm, 0.25 µm (Agilent Technologies, Inc., CA, USA).

The samples were injected manually (1 µL) in the 10:1 split mode. The oven was programmed as follows: hold at 40 °C for 0.5 min, increase to 195 °C in 25 °C/min increments, then to 205 °C in 3 °C/min and to 230 °C in 8 °C/min increments. Temperature remained at 230 °C for 4 min, then increased to 240 °C for 10 min and to 250 °C for 5 min [55].

### 3.8. Antioxidant Properties Evaluation

#### 3.8.1. Ferric Reducing Antioxidant Power (FRAP) Assay

The reducing capacity of samples was determined in accordance with a modified version of a previously reported method [52]. Briefly, immediately prior to analysis, a FRAP reagent was prepared by mixing acetate buffer (pH 3.6) with 10 mM 2,4,6-tris(2-pyridyl)-s-triazine (TPTZ) solution and adding 20 mM of ferric chloride aqueous solution (10:1:1). Then, the mixture was placed into a water bath with temperature set at 37 °C, then 30 μL of each extract diluted in methanol was pipetted into a 96-well microplate, and 180 μL of FRAP reagent was added. After 30 min incubation in dark at 37 °C, the absorbance of the resulting solution was measured at 593 nm wavelength, and the reducing capacity was determined against a FeSO_4_ standard calibration curve. The results are expressed as mmol Fe^2+^/kg of dry material.

#### 3.8.2. DPPH Radical Scavenging Assay

The DPPH assay used for the evaluation of the radical scavenging activity was performed in accordance with the method described in [52]. In particular, 30 μL of each methanolic extract was placed into a 96-well microplate, and 175 µL of a 0.1 M DPPH radical solution in methanol was added. After incubating the mixture for 40 min at room temperature, the absorbance was measured at 515 nm wavelength, and the antioxidant activity was determined against a Trolox calibration standard curve. The results are expressed as mg Trolox equivalents (TE)/kg of dry material.

### 3.9. Statistical Analysis

The statistical analyses were conducted using the R-program (www.R-project.org, accessed on 7 April 2023) [56]. Hierarchical cluster analysis was applied to compute the Euclidean distances between the genotypes and descriptors, employing the Ward’s method. Additionally, Principal Component Analysis (PCA) was performed to reduce the dimensionality of the original variables and transform them into orthogonal variables known as principal components. This analysis facilitated the identification of patterns, relationships and variations within the data.

Microsoft Office 365 was used for the statistical analysis of the TTC, TPC and antioxidant assays. The results are presented as mean value ± standard deviation (SD) of experiments performed in triplicate. For all calculations performed in this work, we used the Durbin–Watson (DW) statistical tests for the residuals, the one-way analysis of variance (ANOVA) followed by the Bonferroni multiple comparisons test for the determination of the respective *p*-values correlations. The latter were determined using the Stata17 program (StataCrop LLC, College Station, TX, USA), and the results of *p*-value correlations are included in this paper as Appendix A. 

## 4. Conclusions

The *V. sativa* plant plays an important role in livestock nutrition, especially in periods of animal food scarcity [35]. Its utilization in rations of monogastric animals, such as pigs and chickens, is closely connected with the control of the vetch’s quantity, since it is known to contain several neurotoxic compounds, such as γ-glutamyl-β-cyano-alanine (GCA) and β-cyano-l-alanine (BCA), which are toxic for their organisms [57]. On the other hand, it is an abundant source of unsaturated fatty acids, especially linoleic, oleic and linolenic acids, and antioxidants, such as isoquercetin, isoliquiritigenin, secoisolariciresinol and chlorogenic acid.

In this study, a substantial variation in the feed quality traits was determined for the tested *Vicia sativa* varieties, highlighting the crucial role and influence the genotype plays in the achievement of high-quality livestock nutrition. Among the six varieties tested in this study, the *Istros* and *M-6900* varieties displayed a particularly intriguing phytochemical profile characterized by high phenolic content, significant antioxidant potency and remarkably high fatty acid indices, highlighting them as perfect candidates for further investigation in the framework of genetic or breeding research. These findings are also indicative of the great potential of these varieties to act as suitable candidates for utilization in farm animal diets either as dry biomass (hay) or as grain in animal feed. 

## Figures and Tables

**Figure 1 plants-12-02807-f001:**
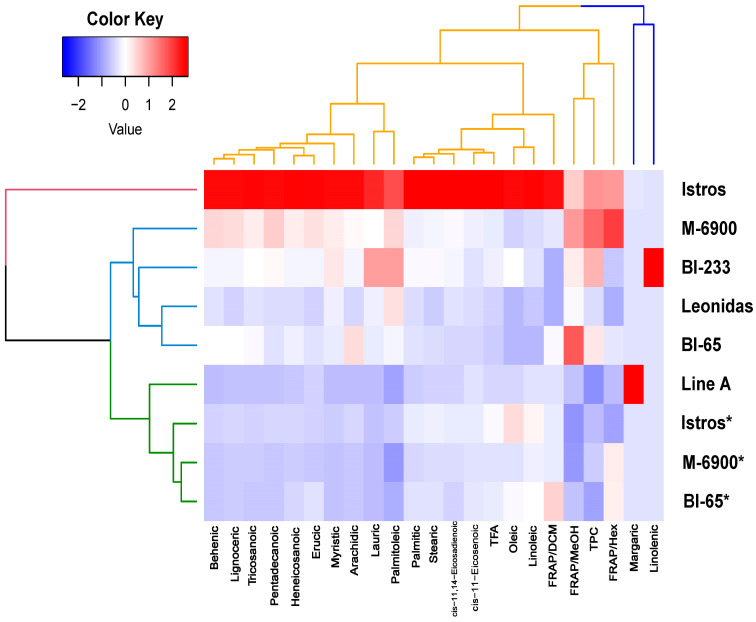
Heatmaps generated for each of the six *Vicia sativa* varieties, illustrating 23 characteristics related to phytochemicals and antioxidants. The color-coded scale is used to indicate an increase (red) and reduction (blue). For the varieties denoted by an asterisk (*), grain was used in the analysis, and biomass was also used for all six varieties. The quality traits and genotypes were subjected to hierarchical cluster analysis using the Ward’s method, and the resulting dendrograms visually display their relationships.

**Figure 2 plants-12-02807-f002:**
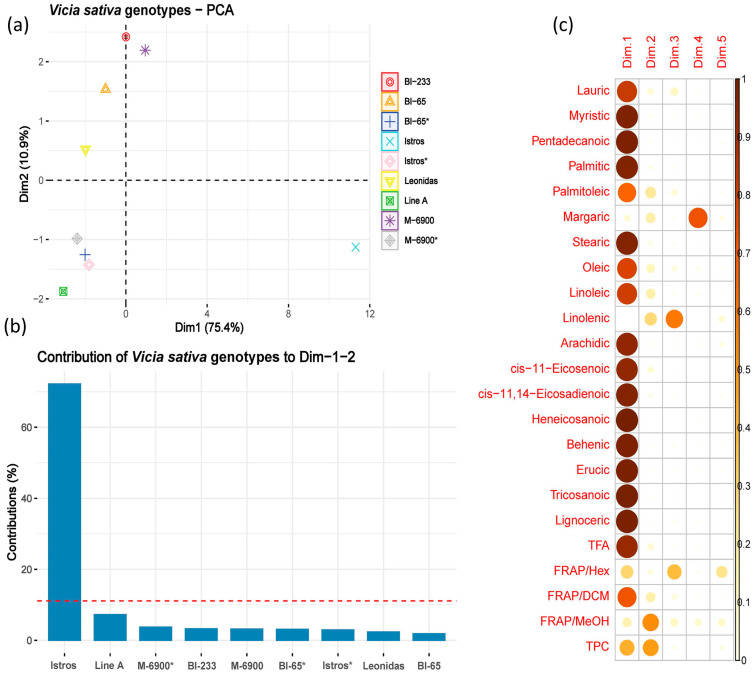
PCA results visualized graphically. (**a**) A plot with the percentage of the variation explained by the first two principal components (86.3%) and the contribution of the varieties to each dimension. (**b**) A bar plot depicting the contribution of the *Vicia sativa* varieties to the first two principal components. (**c**) A correlation plot that highlights the most contributing variables for each dimension. For the varieties denoted by an asterisk (*), grain was used in the analysis, and biomass was also used for all six varieties.

**Table 1 plants-12-02807-t001:** Extraction yields.

Samples	Weight (g)	Yield (%)		
**Biomass**		**Hexane**	**DCM**	**MeOH**
*Line A*	30	0.50	0.85	1.82
*BI-65*	100	0.60	0.69	7.71
*BI-233*	30	0.97	1.61	11.10
*Leonidas*	100	0.45	0.43	5.78
*Istros*	30	2.83	1.37	9.17
*M-6900*	30	0.84	0.57	5.29
Grains				
*BI-65*	30	0.46	0.92	3.75
*Istros*	30	0.56	0.50	1.33
*M-6900*	30	0.49	0.84	3.52

**Table 2 plants-12-02807-t002:** Total tannin (TTC) and phenolic (TPC) content in investigated *Vicia* samples.

Samples	TTC (mg CE/Kg DM) ^1^	TPC (mg GAE/Kg DM) ^2^
**Biomass**		
*Line A*	680.5 ± 12.7	306.4 ± 17.5
*BI-65*	408.7 ± 54.4	1129.3 ± 5.9
*BI-233*	<LOD ^3^	1418.3 ± 22.2
*Leonidas*	<LOD	764.1 ± 6.8
*Istros*	<LOD	1610.2 ± 23.7
*M-6900*	876.3 ± 50.9	1851.3 ± 81.2
Grains		
*BI-65*	1603.2 ± 226.7	420.2 ± 3.1
*Istros*	<LOD	582.2 ± 102.5
*M-6900*	<LOD	656.4 ± 53.3

^1^ TTC (mg catechin equivalent/kg of dry material), ^2^ TPC (mg gallic acid equivalent/kg of dry material), ^3^ <LOD: below the limit of detection.

**Table 3 plants-12-02807-t003:** Fatty acid content in *Vicia* samples (mg/Kg DM).

Compound	*Vicia sativa* (Plant Biomass)						*Vicia sativa* (Grains)		
	*Line A*	*BI-65*	*BI-233*	*Leonidas*	*Istros*	*M-6900*	*BI-65*	*Istros*	*M-6900*
Caproic acid	0.22	1.02	1.56	0.32	5.65	nf	nf	nf	nf
Caprylic acid	0.24	1.26	2.50	1.92	8.45	1.87	0.19	nf	nf
Capric acid	nf	1.73	3.07	2.16	10.22	2.71	nf	nf	nf
Lauric acid	0.27	7.95	28.70	8.95	47.49	11.27	0.37	0.55	0.28
Tridecanoic acid	0.04	0.10	2.00	0.20	0.98	0.22	nf	0.19	0.14
Myristic acid	4.82	16.29	28.80	16.77	83.01	27.83	6.00	8.26	6.24
Pentadecanoic acid	1.77	3.61	5.68	3.21	18.40	8.23	2.08	2.83	2.16
*cis*-10-Pentadecenoic acid	nf	5.97	nf	nf	nf	nf	nf	nf	nf
Palmitic acid	148.72	207.55	287.01	193.30	1052.64	262.07	204.56	239.74	175.32
Palmitoleic acid	0.92	3.77	7.54	5.36	10.22	5.68	1.21	2.23	0.44
Margaric acid	1.025	3.57	5.01	3.45	16.68	5.43	1.27	2.28	1.48
*cis*-10-Heptadecenoic acid	nf	nf	2.35	0.93	nf	nf	nf	nf	nf
Stearic acid	43.04	51.89	77.74	40.77	278.56	74.23	56.07	69.61	52.52
Oleic acid	188.80	125.40	267.18	131.52	719.16	178.48	258.06	342.60	206.94
Linoleic acid	606.72	289.74	607.87	386.52	2691.00	538.82	819.11	931.47	728.91
Linolenic acid	151.49	193.91	201.967	178.07	1004.01	253.02	201.27	247.38	186.83
Arachidic acid	26.04	96.57	63.23	43.38	221.85	76.27	34.80	42.32	32.22
*cis*-11-Eicosenoic acid	6.15	2.95	6.97	3.95	130.23	13.27	8.03	10.32	7.17
*cis*-11,14-Eicosadienoic acid	1.59	1.63	2.75	2.10	11.29	2.90	1.55	2.35	1.88
Heneicosanoic acid	3.57	8.84	9.95	6.31	44.45	14.81	5.88	5.61	4.13
*cis*-11,14,17-Eicosatrienoic acid	nf	nf	10.15	nf	nf	nf	nf	nf	nf
*cis*-5,8,11,14,17-Eicosapentaenoic acid	nf	nf	0.30	1.05	nf	nf	nf	nf	nf
Behenic acid	5.59	20.70	18.12	12.98	76.05	30.71	7.62	9.66	7.37
Erucic acid	8.42	10.96	15.10	8.41	62.79	23.37	10.51	7.51	6.25
Tricosanoic acid	2.38	10.54	10.84	7.04	45.64	14.82	3.36	4.30	3.64
Lignoceric acid	8.69	29.23	24.68	14.91	105.26	42.22	12.11	15.76	12.76
Nervonic acid	0.51	nf	75.60	33.99	228.40	nf	nf	3.82	0.47
Total fatty acids	1211.04	1095.17	1766.68	1107.59	6872.45	1588.24	1634.05	1948.80	1437.15

nf: not found.

**Table 4 plants-12-02807-t004:** Content of phenolic compounds in *Vicia* samples (mg/Kg DM).

Compound	*Vicia sativa* (Plant Biomass)						*Vicia sativa* Grains		
	*Line A*	*BI-65*	*BI-233*	*Leonidas*	*Istros*	*M-6900*	*BI-65*	*Istros*	*M-6900*
Gallocatechin	nf	nf	nf	nf	0.08	0.23	nf	nf	nf
Procyanidin B1	nf	0.11	nf	0.07	nf	nf	nf	0.03	0.02
Chlorogenic Acid	0.02	3.01	0.80	2.33	nf	0.78	0.07	nf	nf
Procyanidin B2	nf	nf	nf	0.04	nf	nf	nf	nf	nf
Daidzin	nf	nf	tr	nf	nf	nf	nf	nf	nf
Quercetagetin-7-o-glucoside	tr	0.03	nf	0.01	0.01	0.01	nf	nf	nf
Epigallocatechin Gallate	nf	nf	nf	nf	nf	nf	nf	tr	nf
Glycitin	nf	nf	tr	tr	nf	nf	nf	nf	nf
Rutin	0.37	11.19	nf	6.22	2.04	5.52	nf	nf	nf
Isoquercetin	6.27	33.91	3.32	9.21	19.95	10.51	tr	tr	tr
Diosmin	nf	0.28	nf	nf	0.23	0.24	nf	nf	nf
Hesperidin	nf	nf	nf	nf	0.13	nf	nf	nf	nf
Quercitrin	nf	nf	nf	0.02	nf	nf	nf	nf	nf
3′,4′,7-trihydroxyisoflavone	nf	nf	nf	nf	nf	tr	nf	nf	nf
Secoisolariciresinol	2.52	11.21	29.82	12.31	46.33	nf	nf	nf	nf
Ononin	nf	tr	nf	nf	0.01	nf	nf	nf	nf
Daidzein	nf	nf	nf	nf	nf	0.01	nf	nf	nf
Glycitein	nf	0.04	0.30	0.10	0.12	0.65	nf	nf	nf
Luteolin	nf	0.54	nf	0.24	0.41	1.82	nf	nf	nf
Apigenin	nf	2.65	0.38	0.37	0.78	2.35	0.01	nf	nf
Genistein	0.04	2.57	1.93	nf	3.86	7.56	nf	nf	nf
Diosmetin	tr	nf	0.16	0.12	0.11	0.24	tr	tr	nf
Isoliquiritigenin	0.01	0.46	0.52	0.13	0.41	1.72	nf	nf	nf

nf: not found; tr: traces.

**Table 5 plants-12-02807-t005:** Results of the DPPH and FRAP antioxidant assays for the investigated *Vicia* varieties.

Samples	DPPH (mg TE/Kg DM) ^1^			FRAP (mmol Fe (II)/Kg DM) ^2^		
	Hex	DCM	MeOH	Hex	DCM	MeOH
**Biomass**						
*Line A*	<LOD ^3^	<LOD	272.1 ± 2.5	3.03 ± 0.04	7.79 ± 0.05	5.82 ± 0.11
*BI-65*	64.2 ± 1.9	<LOD	611.3 ± 31.8	4.78 ± 0.07	10.50 ± 0.20	25.02 ± 0.49
*BI-233*	5.9 ± 7.7	<LOD	<LOD	3.39 ± 0.06	2.70 ± 0.01	13.23 ± 0.02
*Leonidas*	<LOD	<LOD	171.1 ± 69.7	2.25 ± 0.07	2.74 ± 0.04	10.65 ± 0.08
*Istros*	181.3 ± 11.1	<LOD	129.2 ± 46.1	10.81 ± 0.40	34.02 ± 0.70	15.50 ± 0.21
*M-6900*	231.3 ± 34.6	<LOD	1609.9 ± 182.5	14.50 ± 0.29	8.42 ± 0.11	20.00 ± 0.32
Grains						
*BI-65*	<LOD	<LOD	152.3 ± 23.6	6.79 ± 0.03	15.99 ± 0.08	5.70 ± 0.23
*Istros*	<LOD	<LOD	85.0 ± 26.6	1.99 ± 0.03	8.77 ± 0.03	1.92 ± 0.01
*M-6900*	<LOD	<LOD	<LOD	7.21 ± 0.04	9.02 ± 0.20	2.37 ± 0.01

^1^ DPPH (mg Trolox equivalent/kg of dry material), ^2^ FRAP (mmol Fe (II) equivalent/kg of dry material), ^3^ <LOD: below the limit of detection.

## Data Availability

The data are contained within the article.

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
