# Peer review of "An Interdisciplinary Assessment of Biochemical and Antioxidant Attributes of Six Greek Vicia sativa L. Varieties"

_plants, 2023, doi:10.3390/plants12152807_

Round 1

Reviewer 1 Report

The present manuscript evaluates the composition of 6 vicia sativa varieties

The intruduction is complete even some sentences should be completed to highlight the impact of the researche

The results are not described at all, you have nearly just the figures with no description of the data, the discussion is correct the material and methods is ok as well as the conclusion.

My major concerns come with the absence of descrived results just the figures and tables that you have to descript by your self.

Then the fact that there was only one sample taken by variety on the field not showing the potential variability that we could find on the field.

Then there are other minor points to be corrected or discussed:

Line 19 : change potentials by potential 

Line 21 : animal feed is redundant here after animal diets

Lines 31-34; it is not clear in which “substrate” you consider there are not enough references. In Vetch?

Lines 44-48, again, in which ‘substrate’ there are not enough reports? Because it is not in the methodology of definition on nutraceutical that you are working. Please complete correct these you sentences

Lines 48-51; again you need to precise that is on vetch I suppose, if not is too large

Line 52: what you been by variety? Within vetch varieties, or withing all the flora, thus talking about species.. I think is the second

Line 64-65: did you test several environments to test this variability? I think no, so you can express it like this

Line 69: you should include a short analysis of the table 1, presenting the advantages and disadvantages of the varieties. Why you did not analyses the grains of all the varieties?

Line 74-75 you should add again few comments on the results present in table 1

Lines 79-80: same thing

Lines 88-89, same thing

Lines 93-94, same thing, it is not to the reader to extract the conclusions on the tables

Lines 102-104, you need to analyse the clustering

Line 221: change to molecules

Line 23 and 232: change expect by except

Line 310 for used as feed: consider changing to “to be used as feed”

Line 310: you consider that there is no variability in the field, and you took only a sample by variety?

Line 327: change expect by except

Line 339; It’s not clear which quantity you used for the extraction, what is the correspondence to “one part” you weighted the samples precisely?  then what mean 7 parts? The extraction with solvents was made one after the other on the same sample or conducted in parallel.

Line 350: modification or modifications? After dot, please capital letter in briefly

Line 392: please suppress second “method”

Lines 445: why you did not tase the content on  γ-glutamyl-β-cyano-alanine or  β-cyano-l-alanine ?

Globally I recommend to highlight more that the work mainly studied the biomass composition and a few comparisons were made on/with grains.

some sentences are not clear enough in the discussion

Author Response

Please see attached letter

Reviewer 2 Report

The MS deals with the phytochemistry analysis of six Greek Vicia sativa L. varieties.

The topic is interesting and the approach of the Authors deserves the publication. In particular, the cluster and PCA analysis is very relevant and the very fine phytochemical fingerprinting gives a high value to the work.

However, some issues are present in the MS and must be dealt prior to acceptance.

Data reported in Tables 1-4 do not contain statistical analysis (neither descriptive or parametric). These must be both implemented. In particular a lot of discussion in the MS needs to be confirmed by statistical analysis in order to give statistical significance to the differences that the Authors note for instance in section 3.

This is a very important issues: without it several expressions used in the MS (i.e. “most effective” or “more active” or “higher”, “lower”) have no scientific validation.

This must be solved.

In my opinion, the splitting of Results and Discussion sections does not help the reader of the MS. I suggest to merge these two sections.

MINOR POINT:

Lines 112: scientific names should be italicized. Same error repeated several other times along the MS: please, correct.

Author Response

Please see attached letter

Round 2

Reviewer 1 Report

The authors have made all the necessary corrections necessary to now publish the present document.

Author Response

I would like to thank you for accepting the revised version of our Manuscript

Reviewer 2 Report

The Authors effectively addressed the main criticisms dunring R1. 

However, I think that the statistical analysis should be completed in tables 2 and 5 with a non parametric test (i.e. One–way ANOVA followed by the Bonferroni Multiple Comparisons Test) to compare the varieties.

Author Response

Thank you for your suggestion. We have included in the revised version of our manuscript the results of the respective Bonferroni tests